# Piezoelectric Materials: Properties, Advancements, and Design Strategies for High-Temperature Applications

**DOI:** 10.3390/nano12071171

**Published:** 2022-04-01

**Authors:** Yanfang Meng, Genqiang Chen, Maoyong Huang

**Affiliations:** 1State Key Laboratory of Advanced Optical Communications System and Networks, School of Electronics Engineering and Computer Science, Peking University, Beijing 100871, China; 2Center of Flexible Electronics Technology, Tsinghua University, Beijing 100084, China; 3Environmental Simulation and Pollution Control State Key Joint Laboratory, State Environmental Protection Key Laboratory of Microorganism Application and Risk Control (SMARC), School of Environment, Tsinghua University, Beijing 100084, China; genqiangchen@tsinghua.edu.cn; 4Key Laboratory of Photochemical Conversion and Optoelectronic Materials, Technical Institute of Physics and Chemistry, University of Chinese Academy of Sciences, The Chinese Academy of Sciences, Beijing 100190, China

**Keywords:** piezoelectrics, high temperature, materials

## Abstract

Piezoelectronics, as an efficient approach for energy conversion and sensing, have a far-reaching influence on energy harvesting, precise instruments, sensing, health monitoring and so on. A majority of the previous works on piezoelectronics concentrated on the materials that are applied at close to room temperatures. However, there is inadequate research on the materials for high-temperature piezoelectric applications, yet they also have important applications in the critical equipment of aeroengines and nuclear reactors in harsh and high-temperature conditions. In this review, we briefly introduce fundamental knowledge about the piezoelectric effect, and emphatically elucidate high-temperature piezoelectrics, involving: the typical piezoelectric materials operated in high temperatures, and the applications, limiting factors, prospects and challenges of piezoelectricity at high temperatures.

## 1. Introduction

Ever since the discovery of the piezoelectric phenomenon in 1912 [1], piezoelectronics have been generally established and attracted increasingly extensive attention. The piezoelectric effect is that, upon an external load being posed on an object, electrical potential generates on its surface. Piezoelectric materials can serve as crucial units for energy-harvesting equipment or as active parts of sensors, and so on [2]. 

Boosting the output of piezoelectricity, improving the sensitivity of piezoelectric-based sensors and extending its utilization scope are the long-term goals worth pursuing for researchers studying piezoelectronics academically and practically. Besides, some piezoelectric materials in sensors or actuators unavoidably work in elevated-temperature environments (e.g., energy harvestings, the aviation, aerospace and automobile industries and geological explorations) [3,4,5,6,7]. Therefore, it is imperative to explore high-temperature piezoelectric materials to fulfill the aforementioned requirements for their application. However, there are yet only rare reports on high-temperature piezoelectric materials. The piezoelectric materials for elevated application involved: Aurivillius compounds with a layer structure (e.g., Bi_4_Ti_3_O_12_ and related materials), the perovskite BiFeO_3_, quartz and compounds related to the quartz structure, nonferroelectrics, rare-earth oxyborates and nanocomposites. 

In this manuscript, we briefly offer elementary knowledge on the piezotronic and piezophototronic effects and introduce piezoelectrics for high-temperature applications (Section 3, Section 4, Section 5, Section 6 and Section 7) in detail. Section 2 illustrates the basic knowledge on the piezotronic and piezophototronic effects. Section 3 simply introduces the mechanism of piezoelectricity at high temperatures. Section 4 outlines what the limiting factors are in using piezoelectricity at high temperatures. Three aspects—the Curie temperature, conductivity and chemical stability—and the dielectric properties regarding the performance and application of piezoelectric materials at high temperatures are comprehensively analyzed. Section 5 reviews the main classes of high-temperature materials. Furthermore, how the above materials meet the limiting factors in Section 4 is elucidated. Section 6 offers insights for the designation of high-temperature piezoelectric materials. Finally, we summarize the findings and propose challenges and prospects for further study of piezoelectric materials in high-temperature applications.

## 2. Fundamentals of the Piezoelectric Effect

### 2.1. Mechanism of the Piezoelectric Effect

The common characteristic of piezoelectric materials is that their crystalline structures contain crystal lattices that lack center symmetry [8,9]. Piezoelectric properties include the direct piezoelectric effect and the reversed piezoelectric effect. The direct piezoelectric effect refers to the phenomenon where, under an external load, polarization produced by the inner center of positive charges and the center of negative charges undergoes displacement, and the reverse signs of the charges are produced at the two ends of dielectric surfaces. Under an external force (within a certain range), the charge density of the piezoelectric material is proportional to the external force, which can be defined as: δ=dT

δ is the facial charge density, *d* is the piezoelectric strain constant and *T* is stretching stress. On the contrary, the reversed piezoelectric effect is denoted as when an electrical field is imposed on piezoelectric materials, and the inner center of positive charges and the center of negative charges undergo displacement. Consequently, polarization occurs and generates a strain on the material. As the magnitude of the electrical field is within a defined range, the magnitude of stress and the output electrical signal turn out to have a linear relationship: *x* = *dE*.

*x* is the strain, *E* is defined as the electric field and *d* is assigned as the reverse piezoelectric strain constant.

The magnitude of the piezoelectric energy can reflect the coupling constants of the elastic properties and the dielectric properties, defined as the electromechanical coupling coefficient, which can be expressed as follows:(1)$$K=\sqrt{\frac{u_{12}^{2}}{u_{1}u_{2}}}$$

*u*_12_ is defined as the piezoelectric energy, *u*_1_ is assigned as the elastic energy and *u*_2_ is defined as the dielectric energy.

Under an electrical field, the electrical displacement *D* of non-piezoelectric materials under a free-standing state (no external force) is given as: *D* = *εE*, where *ε* is assigned as the dielectric coefficient of the dielectric medium. In the absence of an electrical field, when stress *T* is imposed upon the non-piezoelectric materials, strain *S* is induced. The relationship is shown as: *S* = *sT*, where *s* is assigned as a flexible constant.

The relationship between the electrical and mechanical straining of piezoelectric materials can be described as *D* = *dT* + *εE* (positive piezoelectric effect)
*S* = *s^E^T* + *dE* (negative piezoelectric effect)

Taking ZnO as an example [10] (as shown in Figure 1a), the 2*p* orbital of Zn cations and O^2−^ anions are included in a crystal lattice. In a free-standing state, due to the overlapping between the center of the positive charges Zn^2+^ and the center of the negative charges O^2−^, the dipole moment is zero. If an external force is imposed on the crystal lattice of ZnO, the dipole moment is produced by the relative displacements of the centers of the O^2−^ and the Zn^2+^. The dipole moments of every crystal lattice bring about electric potential at the macroscopic level, which can be defined as piezoelectric potential (piezopotential) (as shown in Figure 1a).

### 2.2. Piezotronics

When imposed upon by an external force at the microscopic level, piezoelectric potential (piezopotential) is established within the inner structure of the materials, with the surface charges redistributed and the polarization phenomenon produced. As two electrodes sandwich this piezoelectric material, a piezopotential-driving transient current of electrons is produced under the external load. With the development of electronic devices triggered or controlled by strains (e.g., force, pressure) and sensors based on the piezoelectric effect, an edge-cutting discipline has sprung up—Piezotronics [11,12].

Effect field transistors (FET) are an important representation of piezotronic devices. As shown in Figure 1b [13], a constant voltage is applied to the two ends of piezoelectric nanowire material as a source-drain voltage. Piezoelectric potential is generated via the external-pressure-induced deformation of nanowire (a tensile strain in the upper panel of Figure 1b, and a compress strain in the bottom panel of Figure 1b) and serves as a gate voltage. That is, piezoelectric potential that is applied to the channel of FET is equivalent to a gate voltage applied to the channel of FET. The piezoelectric potential changes the Schottky barrier at the source side. As shown in Figure 1c, the SBH at the source side reduces from φ_s_ to φ_s_′ ≈ φ_s_ − ΔE_P_ in the tensile deformation (outward deformation) state. As shown in Figure 1d, the SBH at the source side raises from φ_s_ to φ_s_″ ≈ φ_s_ + ΔE_P_′ in the compressive deformation (outward deformation) state.

### 2.3. Piezophototronics

Piezophototronics is an interdisciplinary subset of piezoelectricity, photonic excitation and the science of semiconductors. Piezophototronics is the discipline of investigating the devices which work through tailoring or controlling the electro-optical process by strain-induced piezoelectric potential. Piezophototronics has gained a wide application in photoelectric devices (e.g., light-emitting diodes, photocells and solar cells) owing to its improvements to the performances of photoelectric devices [14,15]. The candidate for piezotronics and piezophototronics is mainly the wurtzite family [16,17]. To guarantee the balanced injection of heterogeneous junctions, which is heavily related to the efficiency of a LED, the piezophototronic effect can be employed to address the issue of the difficulty (due to band alignment and various material properties) of injecting an equal amount of electrons and holes into the space-charge region. For instance, Yang et al. [16] constructed a hybrid LED using an n-type ZnO and a p-type PEDOT:PSS (as show in Figure 1e). As shown in Figure 1e, the PEDOT:PSS coated half of the ZnO NW on a flexible substrate with separated gold electrode pads. As the substrate bended, ZnO NW generated piezopotential. The middle and bottom panel of Figure 1e present the simulated results of the distribution of the piezopotential. Figure 1f illustrates the experimental results of the *I-V* curve, suggesting that positive and negative piezocharges can be induced at the respective interfaces of the heterojunction. 

## 3. Piezoelectricity at High Temperatures

Most materials for piezotronics or piezophototronics work around room temperature. However, some piezoelectric materials for sensors or actuators are adopted in elevated temperature environments, such as in applications for automobiles, aerospace, aviation, energy, geological exploration and so on. Therefore, it is highly desirable to explore more of these special piezoelectric materials. 

### 3.1. The Principle of Piezoelectricity at High Temperatures

The symmetry of a crystal is determined by its lattice, which is strictly bounded by the regulation of the inner structure of the crystal and the brave protocell, which involves the *Emnoman* principle. The *Emnoman* principle, [18,19] states that the physical symmetry of any crystal are included with the symmetry of its crystal point group. When the crystal experiences deformation under a mechanical force, the charged positive ions or negative ions of the inner crystal are displaced, which causes the overall electrical dipole moments to change. In each type of point group with center symmetry, the arrangements of the symmetry centers of the ions are not susceptible to being destroyed, and therefore the centers of the ions are not displaced. If the point group of the crystal has hetero-pole center symmetry in absence of homo-pole center symmetry, the overall electrical dipole moments of the crystal change upon the application of the mechanical force. 

The piezoelectric crystal contains a pyroelectric crystal with its surface charged by temperatures varied in a uniform style, indicating that spontaneous polarization occurs in the crystal. When the crystal experiences thermal expanding, the relative displacements of the positive and negative ions change the spontaneous polarization in a unique pole axis of the crystal. In regard to pyroelectric crystals, some crystals only experience spontaneous polarization in specific temperature ranges, and the external electrical field can reverse the intensity of the spontaneous polarization, making the crystal ferroelecric.

Pyroelectric crystals can be regarded as a thermodynamic system, with unified relationships between its thermal parameters, mechanical parameters and electrical parameters. The system is in a thermal isolation state with no entropy change due to the fast conversion rate between the mechanical parameters and electrical parameters to overlapped thermal exchanges. 

### 3.2. The Factors of the Piezoelectric Nanocomposites Required to Be Considered for Application at High Temperatures

High-temperature sensors are a core part of high-temperature, mechanically controlled systems. The stability of the sensors that are applied in high temperatures is closely related to the properties of piezoelectric materials.
(1)Phase transition may make the properties of the materials fluctuate as the temperature changes.(2)Resistivity interferes with the migration of surficial changes induced by the piezoelectric effect.(3)Thermal stability.(4)The acoustic wave decays in high temperatures and with dielectric losses.(5)Chemical stability.(6)The component decomposition and the lattice-defect formation [20,21,22].

Among the above six factors, the most significant factor is the phase transition. For example, GaPO_4_ is a kind of high-temperature piezoelectric crystal material with outstanding properties: high resistivity and low dielectric loss. However, the crystal undergoes phase transition near 930 °C, limiting its application in high temperatures. As another example, the piezoelectric coefficient of AlN is desirable and has no phase transition near 930 °C. Nevertheless, the undesired crystal defects (stemming from a physical gas-phase growth procedure) also decreases its resistivity and limits its upper-limitation temperature. For instance, high resistivity and high RC time constants are required for high-temperature sensors. Higher RC time constants give rise to a longer time period for the charges to stay in the sensor. The longer retention time of the charges is feasible to be detected. Moreover, the minimum effective frequency heavily depends on the RC time constant. The lower frequency of the sensor can be defined as
(2)$$f_{LL}=\frac{1}{2\pi \mathrm{RC}}$$

The characteristic angular frequency ω = 1/(RC) and ω = 2*πf*, so *f**_LL_*= 1/(2*π*RC).

Below this value, the induced charges may vanish before being detected. Higher RC time constants may facilitate the sensors operating in lower frequencies. With respect to LN crystals, though the electromechanical coupling coefficient is greater and the *T*c reaches 1170 °C, the resistivity of the materials is relatively lower, making them usable only in temperatures below 600 °C [20]. Besides phase transition and resistivity, thermal stability, the decaying of the acoustic wave in high temperatures, dielectric loss and chemical stability are also important factors that are supposed to be given emphasis.

## 4. What Are the Limiting Factors in Using Piezoelectric Materials at High Temperatures?

The number of dielectric constants is associated with symmetry. A high symmetry is associated with smaller dielectric constants, which are the smallest for triclinic systems with high symmetry. A triclinic system has six independent dielectric constants: *ε*_11_, *ε*_12_, *ε*_13_, *ε*_22_, *ε*_23_ and *ε*_33_. The symmetry of the cub system is highest. The cub system has only one independent dielectric constant: *ε*_11_.

Untreated piezoelectric materials are comprised of anisotropic poly-crystals, and transform to an anisotropic dielectric medium after treatment. The *x* axis and *y* axis are the non-polarization axis and the *xy* plane is the isotropic plane. There is no difference between the *x* axis and *y* axis. However, the *z* axis is totally different. The symmetry of polarization-treated piezoelectric ceramics is equivalent to the symmetry of a cylinder. The *xy* plane is an anisotropic plane rather than a isotropic plane. The most important parameter to characterize the piezoelectric properties of piezoelectric materials is the piezoelectric coefficient: *d_ij_*_._
(3)$$d_{ij}=\frac{N_{s}Z^{*}a}{k_{B}T}\left\{\langle u_{i}\varphi_{H,j}\rangle -\langle u_{i}\rangle \langle \varphi_{H,j}\rangle \right\}$$
where *T* is the temperature, *k_B_* is the Boltzmann constant, *N*s is the total number of five-atom cells composing the supercell and *Z** is the Born effective charge associated with the soft mode and the five-atom lattice constant. *u_i_* is the *i*-component of the supercell average of the local mode at a given *MC* sweep, and *Z_H,j_* is the *j* component of the homogeneous strain tensor (in Voigt notation) at this MC sweep. The *φ_H,j_* symbol denotes statistical averages over the different *MC* sweeps [23]. 

Li et al. reported [24] piezoelectric ceramics with Ba^2+^ substituted for Ca_2_Nb_2_O_7_. The detections from XRD and Raman characterization indicated that the solid-dissolve limitation is 0.3. However, the piezoelectric coefficient was not reported. Gao et al. developed [25] a piezoelectric ceramic of Sr_2–x_Ba_x_Nb_2_O_7_ by SPS with Ba^2+^ doping, increasing *d*_33_ to 3.6 pC/N. Moreover, the non-in-situ annealing experiment suggested the thermal annealing polarization temperature was above 1150 °C. They further prepared the textualized Ce^3+^-doping piezoelectric ceramics with La_2_Ti_2_O_7_ and successfully promoted *d*_33_ to 3.9 pC/N.

The *d_ij_* is a parameter from the macro-scope viewpoint, but on other hand, from the micro-scope viewpoint, the theoretical expression of the total internal energy is effectively *Hamiltonian* and contains a local-mode self-energy, a long-range dipole–dipole interaction, a short-range interaction between soft modes, an elastic energy and an interaction between the local modes and local strains [26]. In particular, the local-mode self-energy is given by:(4)$$E_{tot}=\sum \left\{k_{2}u_{i}^{2}+\alpha u_{i}^{4}+\gamma \left(u_{i,x}^{2}u_{i,y}^{2}+u_{i,y}^{2}u_{i,z}^{2}+u_{i,z}^{2}u_{i,x}^{2}\right)\right\}$$

Here, the sum runs over all the <B> sites, and (*u_i,x_*, *u_i,y_*, *u_i,z_*) are the Cartesian components of *u_i_* in the orthonormal basis formed by the [100], [010] and [001] pseudo-cubic directions [27].

### 4.1. Curie Temperature

#### 4.1.1. The Importance of the Curie Temperature

The Curie temperature refers to the temperature at which the spontaneous magnetizations of magnetic nanocomposite materials decrease to zero, and is the critical point at which ferromagnetic or ferromagnetic nanocomposite materials become paramagnetic. The Curie temperature (*T*_C_) determines whether piezoelectric materials can apply to high-temperature devices due to the fact that, above the Curie temperature, most piezoelectric materials for high-temperature application lose their piezoelectricity, providing a non-trivial limitation for their utilization in relatively low temperatures.

It is widely acknowledged that PZT has advantages including a large piezoelectric coefficient, high piezoelectric polarizability, excellent stability and low cost. However, the loss of piezoelectric characteristics at the Curie temperature (*T*_C_) limits their application at high temperatures, which is the untrivial problem for its application in occasions of elevated temperatures. PVDF exhibits very low thermal stability (melting at 177 °C), let alone being able to maintain stability at its Curie temperature, suggesting its unsuitability for operating in high temperatures. A high Curie temperature is the prerequisite of great significance for piezoelectric nanocomposites in high-temperature applications. For example, among the large quantity of high-temperature piezoelectric materials, bismuth layer-structured ferroelectrics [28] (BLSFs, also called Aurivillius phase compounds) have been the most appealing material for applications in piezoelectric sensors at high temperatures, owing to its incomparably high Curie temperature

The Curie temperature has a profound influence on the performance of piezoelectric materials. It is worth noting that the Curie temperature is strongly related to de-polarization, which may occur when the practical operate temperature is much lower than the Curie temperature. Furthermore, both the Curie temperature and the de-polarization temperature determine the highest temperature at which piezoelectric materials can be used. As an example, Liao et al. [29] have reported Bi_2_WO_6_, as the simplest member of the Aurivillius-based piezoelectric materials, and improved it to possess an ultra-high depolarization temperature and an ultra-high Curie temperature. 

#### 4.1.2. The Effect on the Curie Temperature of Piezoelectric Composites for High-Temperature Applications

(a)The instinct of materials

Among the piezoelectric materials that possess a high Curie temperature, perovskite layer-structured (PLS) piezoelectric ceramics have gained increasing interest due to its good thermal stability [30]. 

With its unparalleled high Curie temperature, PLS piezoelectric ceramics have become the hot topics in the field of high-temperature piezoelectric ceramics. (e.g., Sr_2_Nb_2_O_7_ and Pr_2_Ti_2_O_7_ possess the Curie temperature of 1342 °C and 1555 °C [31]). The PLS piezoelectric ceramics also possess ultrahigh electrical resistivity (higher than bismuth layer-structured piezoelectric ceramic, perovskite and LiNbO_3_ single crystals by two-to-four orders of magnitude [32,33]).

(b)Inner structure

The high Curie temperature of piezoelectric materials is closely related to their inner structure. Wu et al. [34] developed high-temperature piezoelectricity materials from the (1−x−y)K_1-w_NawNb_1−z_Sb_z_O_3-x_BiFeO_3_−yBi_0.5_Na_0.5_ZrO_3_ (KNwNS_z-x_BF_-y_BNZ) ceramics. The most outstanding characteristic of this material is its extremely high Curie temperature, which is realized by optimizing *x*, *y*, *z* and *w*. The characterization indicated that the high *T*_C_ was associated with the phase structure. To be specific, the intimate coexistence of the rhombohedral (*R*) and tetragonal (*T*) phases inside the nanodomains was detected by atomic-resolution polarization and mapped by *Z*-contrast imaging. The high Curie temperature can be explained by the nearly vanishing polarization anisotropy, which in turn lowers the domain wall energy and makes the polarization rotation between different states easier.

In addition, the Curie temperature is also closely related to the amount of the second phase. A small amount of the second phase and the solid solution turns out to create trigonal distortions in a crystal structure and generally disappear as the *x* increase. 

(c)The morphotropic phase boundary (MPB)

A hot topic concerns why the materials in the morphotropic phase boundary (MPB) possess exceptional properties. The MPB is the co-existence of the square syngony domain with the tetragonal syngony domain, and the free energy difference between them is very small. This structure, with regard to spontaneous polarization, can be feasible for experiencing crystallographic change transitions under the external electric field, leading to the field-induced phase transition phenomenon. As a result, the migration and polarization of active ferroelectrics may be dramatically facilitated. Therefore, the dielectric constant and piezoelectric coefficient reach their maximum value. Additionally, due to the MPB facilitating ion migration, the electric domain movements become easier in PZT ceramic with an MPB and the mechanical loss energy increases, leading to lower *Q*m. Noheda et al. [35] made a great breakthrough in the knowledge of MPBs by investigating Pb(Zr_0.52_Ti_0.48_). Their measurements indicated that as the temperature decreased from 736 K to 300 K, the original cubic phase of the crystal became a tetragonal phase, and became a monoclinic phase below 200 K. 

(d)The tolerance factor

The tolerance factor can be employed to predict the Curie temperature. The smaller the tolerance factor, the higher the Curie temperature.

The tolerance factor $t=\frac{r_{A}+r_{0}}{\surd 2\left(r_{B}+r_{0}\right)}$ [36].

The tolerance factor can be used to justify the stability and phase structure, and in a typical cubic perovskite is 1. The range of *t* for a stable perovskite structure is 0.88–1.09. As *t* < 1, trigonal distortion crystals play a dominant role, while as *t* > 1, tetragonal crystals occupy a dominant position. 

MPB is located at the position with *PT* content 64mol% *d*_33_ = 460 pC/N and the electromechanical coupling coefficient *k_p_* = 0.56, equivalent to a PZT possessing a Curie temperature of 450 °C. Eitel et al. [37] predicted the BiInO_3,_ MPB component of a solid-solution complex of BiYbO_3_ and PbTiO_3_ are 550 °C and 650 °C, respectively.

(e)The morphology of piezoelectric materials

Curie temperature is also strongly related to the lattice potential energy of the crystalline properties, and further affects the morphology of the crystalline properties of the piezoelectric materials. Therefore, designing the microstructure of piezoelectric materials is a reasonable method for tailoring their Curie temperatures. However, the comprehensive acquaintances of the ferro-/piezoelectric properties of bismuth scandate–lead titanate (BS–PT) have been dramatically prohibited by a shortage of information, especially regarding microstructures, and are incompatible with the growth of high-quality crystals. Luo et al. [38] explored the microstructure of BT–PT crystals near the morphotropic phase boundary from a microscopic perspective. The multi-level domain structures in the 〈001〉_cub_ BS–PT single crystals were detected by an integration of temperature-variable birefringence imaging microscopy and piezo-response force microscopy, combined with a high-resolution X-ray diffraction in the temperature range of 30–500 °C. The crystals turned out to have temperature-dependent morphologies. Specifically speaking, at room temperature, the morphologies of the crystals showed a tetragonal symmetry, while at the higher temperature of 460 °C, the morphologies transformed to cubic symmetry. These investigations of domains, symmetries and phase transitions offered a deep comprehension of the structure of BS–PT crystals and other related ferro-/piezo-electric single crystals on multiple length scales, and paved the avenue for exploiting new high-*T*_C_ piezoelectric materials.

#### 4.1.3. The Approach for Improving the Curie Temperature of Piezoelectric Nanocomposites for High-Temperature Applications

(a)Doping

Doping is a general method for promoting *T*c. For example, Chen et al. [39] systematically synthesized CaBi_2_Nb_2_O_9_-based high-temperature piezoelectric ceramics using co-doped Mo/Cr, demonstrating that the co-dopant method has effectively improved the Curie temperature (reaching to an ultrahigh *T*c of ~939 °C) and thermal stability. 

As another instance, Guo et al. [40] fabricated high-temperature piezoelectric materials using BiFeO_3_–BaTiO_3_–NdCoO_3_ with multiferroic characteristics and carried out a systematic study on the structural, piezoelectric and multiferroic features of the materials. It was mentioned that the addition of 1.0–3.0 mol% NdCoO_3_ elevated the Curie temperature from 486 °C to 605 °C, which can be explained by the morphotropic phase boundary of the rhombohedral and orthorhombic phases. 

(b)Optimized composition

Besides doping, optimized composition is a common method to increase Curie temperature. As an instance, Guo Q. H. et al. [41] have broadened the range of operation temperatures for Sm-modified 0.15 Pb(Mg_1/3_Nb_2/3_)O_3_-(0.85-x)PbZrO_3_-xPbTiO_3_ by delicately designing the composition.

#### 4.1.4. Improving the Curie Temperature of Piezoelectric Nanocomposites Is Not Supposed to Deteriorate the Other Properties

Admittedly, scientists all make unremitting endeavors to pursue higher Curie temperatures. It is worth mentioning that the improvement of the Curie temperature is not supposed to deteriorate the other properties. On the other hand, the improved other properties besides Curie temperature are not supposed to have a negative impact on Curie temperature. However, the pursuit of higher Curie temperatures has generally deteriorated other performance properties. For example, un-doped perovskite displays a high Curie temperature (*T*c) with shortcomings of a rather low mechanical quality factor (*Q*_m_), which may decrease its energy conversion efficiency. In response to this issue, Wang et al. [42] endeavored to improve the mechanical quality factor (*Q*_m_) of the piezoelectric ceramic BYPT-PZN-xMn by deeply analyzing the microstructure and electric properties of new Mn-doped perovskites without degrading the advantages of high Curie temperatures. The results verified that the Mn addition not only had a negative impact on the Curie temperature of the material,.

There are two shortcomings with respect to the bismuth layer-structured ferroelectric materials: the high coercive field, which does not facilitate polarization, and the undesired piezoelectric properties. To overcome the above drawbacks, procedure modifications and doping modifications are employed. Doping modifications can improve the piezoelectric properties of bismuth layer-structured ferroelectric materials, but at the cost of decreasing their Curie temperature.

#### 4.1.5. Can the High-Temperature Piezoelectric Nanocomposites Operate above Curie Temperature?

Generally, the piezoelectric materials work below Curie temperature. Can the high-temperature piezoelectric nanocomposites operate above Curie temperature? Experimental results from Mbarki et al. [43] offered the answer. They applied continuum modeling to quantitatively analyze the possibility of achieving apparent piezoelectric materials with large and temperature-stable electromechanical coupling in a wide temperature range that extended remarkably above the Curie temperature. The barium and strontium titanate was selected as the experimental materials, and the result showed the possibility for high-temperature piezoelectric materials to work above Curie temperature by electromechanical coupling, and showed temperature stability up to 900 °C.

### 4.2. Conductivity

#### 4.2.1. The Importance of Conductivity

As aforementioned in Section 3.2, high resistivity and high *RC* time constants are basic requirements for a high-temperature sensor. On the one hand, a higher *RC* time constant suggests a longer retention time for the charges to stay in the sensor, facilitating their detection. On the other hand, the *RC* time constant also influences the minimum effective frequency. The lower frequency of a sensor can be expressed as Equation (2).

Furthermore, high conductivity has a negative effect on the pole of the piezoelectric ceramics, while a too low conductivity may degrade the sensitivity of the device for its application to sensors. Therefore, the conductivity is needed to mediate. However, although the piezoelectric materials operated in evaluated temperature can maintain high-temperature stability, their conductivity and deteriorating sensitivity may degrade. For example, with respect to a sensor based on the AlGaN/GaN transistor, the resistance of the channel of the transistor depends strongly on the relative change in pressure and temperature, due to the mobility and carrier concentrations having a function relationship with the temperature. As a result, measurements in high-temperature and high-pressure conditions become impractical [44,45,46]. The inappropriate resistivity also brings about undesirable issues (e.g., leakage currents). For example, due to the volatility of Bi^3+^ of BF–BT, the valence change occurs in Fe^3+^(Fe^3+^ to Fe^2+^) and produces oxygen vacancy, leading to a large undesired leakage current at room temperature and difficulties with polarization. This may be responsible for the rare reporting on this material. Although some high-temperature piezoelectric materials possess unexceptional Curie temperatures (1170 °C), piezoelectric coefficients and electromechanical coupling coefficients, the resistivity of the materials is relatively low at high temperatures, and it can only be used in temperatures below 600 °C.

#### 4.2.2. The Approach to Adjust Conductivity

(a)Suitable dopant substitution

To tailor the conductivity, a suitable dopant substitution is recognized as a general but important approach. However, it needs to be ingeniously designed to obtain piezoelectric ceramics with high conductivity merely by combining the ion doping substitution with a typical sintering procedure. For example, in 2009, Yan et al. [47] prepared Nd_2_Ti_2_O_7_ with a high density (>98%) and high degree of orientation (~0.82) by suitable dopant substitution to raise the *d*_33_ to 2.6 pC/N.

(b)Co-doping

With respect to a high-temperature piezoelectric compound based on oxide, the conductivity strongly depends on the oxygen vacancy concentration. Therefore, the conductivity alteration also can be completed by adjusting the oxygen vacancy concentration. For example, with respect to Bi_4_Ti_3_O_12_ (BIT)-based piezoelectric material, during the sintering process, high oxygen vacancy concentrations generated by the volatilization of the bismuth element pose the great challenge for its practical application. Fortunately, co-doping can alleviate this problem. In Li’s work [48], the conductivity of the BIT ceramic was modulated by tailoring the oxygen vacancy concentration using Cu and Sb co-doping. The experimental results showed that co-doping drastically reduced the oxygen vacancy concentration and enhanced electrical resistivity, as well as improved the piezoelectric properties. 

(c)Adding the second unit

Besides suitable dopant substitution and co-doping, alternatively, adding the second unit is also an important avenue to improve dielectric properties. For one, the process of forming a solid solution of multiple compounds can reduce the free energy and facilitate the sintering process. For another, with the variation of the content of the second unit, the phase structure also changes, giving rises to the morphotropic phase boundary (MPB) or polymorphic phase transition [49]. By incorporating the perovskite NaNbO_3_ into Ca_2_Nb_2_O_7_ and Sr_2_Nb_2_O_7_, Titov et al. designed a novel ferroelectric niobate by means of the evolution of the fractal crystal structure [50]. 

Employing the ultrahigh-temperature piezoelectric material, CaBi_2_Nb_2_O_9_ with the highest Curie temperature *T*c, Long C. B. et al. reported [51] that electrical resistivity of CaBi_2_Nb_2_O_9_ ceramics could be remarkably enhanced by BiMn co-substitution at the A and B positions. 

(d)By modifying dielectric constant

The dielectric constant has had a great influence on the conductivity of piezoelectric materials, and further, a great impact on the piezoelectric properties of piezoelectric materials. Some reports discovered that domain wall motion has untrivial effects on the dielectric constant, giving rise to changing conductivity for piezoelectric materials. The 90° domain wall motion has made a tremendous contribution to the dielectric constant of piezoelectric materials. Arlt et al. [52] had put forward a model that elucidated these contributions phenomenologically to separate the instinct effects from the domain wall’s motion contribution. By means of this model, they discussed variations in the observations of piezoelectric at low and high temperatures and at mediate and high frequencies that contributed to the 90° domain wall’s motion contribution.

In another instance, Li et al. [53] have found that the quenching process had a positive impact on the dielectric, ferroelectric and piezoelectric properties of 0.71 BiFeO_3_–0.29 BaTiO_3_ ceramics with Mn modification (BF–BT xmol% Mn). Yang et al. prepared [54] a high-temperature piezoelectric ceramic of MnO_2_-doped 0.69 BiFeO_3_–0.02 Bi(Mg_1/2_Ti_1/2_)O_3_–0.29BaTiO_3_ (BF69–BMT2–BT29) by combining the strategies of refined electroceramic processing and Mn-doping, and adding a third BMT member to decrease the dielectric-loss of the optimized material. 

### 4.3. Thermal Stability

Thermal stability is the factor that is of the greatest significance for piezoelectric materials operating at high temperatures. One of the methods to improve the thermal stability of the materials is atom substitution. For instance, Pardo et al. [55] prepared a sodium-substituted lithium niobate and systematically investigated the piezoelectric, elastic and dielectric coefficients by impedance measurements at various temperatures. Unexpectedly, the Na content had considerably increased the thermal stability of the material, endowing it with piezoelectricity even at temperatures above 600 °C. This suggested that foreign atom substitution provided a powerful approach for improving the thermal stability of piezoelectric materials. For another example, with respect to the crystal CTGS, Ca_3_NbGa_3_Si_2_O_14_(GNGS) Sr_3_TaGa_3_Si_2_O_14_(STGS) Sr_3_NbGa_3_Si_2_O_14_(STGS), when it turned out to be in a disordered state at high temperatures, the diffusion of oxygen vacancies caused evaluations of its thermal conductivity and increased the loss of the crystal, which deteriorated its performance at high temperatures. The disordered crystals possess a relatively higher resistivity and comparatively more stable electromechanical properties at high temperatures, as well as the *Q*-factor. Taking the cost, the evaluated resistivity and the *Q*-factor into consideration, and for the purpose of decreasing the excess Ca^2+^ ions, Ca^2+^ is replaced by Al^3+^. As a result, the doped optimized crystal is obtained [56].

### 4.4. Chemical Stability

Chemical stability is of great significance for piezoelectric materials operated at high temperatures. Some high-temperature piezoelectric materials (e.g., perovskites) are vulnerable to organic solvents. For long-term use, the chemical stability of high-temperature piezoelectric materials is essential. Therefore, it is highly desired to develop mediated and chemically-stable high-temperature piezoelectric materials. For instance, by using the slow evaporation solution growth technique, Murugan et al. have fabricated [57] piezoelectric materials with l-Arginine 4-nitrophenolate 4-nitrophenol dihydrate single crystals possessing both good chemical stability and fine piezoelectric coefficients, by means of slow evaporation technology. The chemical stability could be ascribed to the fact that the phase of the crystal quality was nearly perfect in absence of structural grain boundaries. 

### 4.5. The Synergy of the Factors

The synergy of the above factors is also of great significance and is expected to be given high priority. It is unwired for pursuing high *T*c at the cost of degrading the chemical stability. Zheng et al. have studied [58] the synergistic relationships among the Curie temperature of piezoelectricity and the stability of potassium–sodium niobhrate piezoelectric ceramics. To solve the problem of improving to a large piezoelectricity (*d*_33_), a high Curie temperature (*T*_C_) and improved temperature stability synergistically for (K,Na)NbO_3_ (KNN)-based ceramics, they modified the chemical composition to tailor both *d*_33_ and *T*_C_ to alter the phase boundaries. The optimized relationships among the composition, phase boundary and electrical properties achieved the synergy and improved these parameters for this high-temperature piezoelectric material. Therefore, this work sheds light on the comprehensive acquaintances of KNN-based ceramics with a high *T*_C_.

Taking different properties into comprehensive consideration, special process methods are also utilized. Dan Yu et al. [59] presented a novel method for thermal gradient sintering to fabricate the high-temperature piezoelectric crystals Bi_12_TiO_20_-BaTiO_3_, and realized improvements for both the piezoelectricity and Curie temperature.

## 5. The High-Temperature Piezoelectric Materials

The continuous development of senor devices, actuators and other precise instruments for high-temperature operations poses new challenge for the development of piezoelectric materials for the automotive and aerospace industries, as well as others [21,60]. 

The generally used piezoelectric materials for high temperatures are non-ferroelectrics (including quartz and compounds related to the quartz structure and rare-earth oxyborates); layer-structured and Aurivillius compounds; the perovskite BiFeO_3_ and its solid solution binaries with PbTiO_3_, BaTiO_3_ and (K_0.5_Bi_0.5_) TiO^3+^; and other ternaries (e.g., all the above). 

### 5.1. The Basis of High-Temperature Piezoelectric Materials

#### 5.1.1. Non-Ferroelectrics

(a)Quartz and compounds related to the quartz structure

The quartz crystal, as one of the most primitive piezoelectric materials, has being extensively utilized owing to its unparalleled stability. The quartz crystal [31,61,62] is a kind of double-crystal and experiences *α-β* phase transition. The unwanted transition may severely deteriorate the overall properties of the crystal. Consequently, the temperature range is considerably altered. The natural mineral tourmaline possesses higher resistivity, but the components of the material are extremely complex, such as (Ca,K,Na) (Al,Fe,Li,Mg,Mn)_3_(Al,Cr,Fe,V)_6_(BO_3_)_3_(Si,Al,B)_6_ O_18_(OH,F)_4_, and impractically obtained by synthesis [63]. The piezoelectric ceramics based on quartz crystals possess a layered structure with a piezoelectric coefficient approaching 14 pC/N. However, it shows a heavy dependency on the *T*c of the materials, limiting its operating temperature to below 800 °C. The quartz has several disadvantages, such as a relatively lower piezoelectric coefficient (2.31 pm/V), a low phase transition temperature (573 °C) between phases, and mechanical twinning at 300 °C [64].

(b)Rare-earth oxyborates

The high-temperature piezoelectric materials based on rare-earth oxyborates have been rarely reported, because the rare-earth elements are uncommon.

With the development of electronics and sensing technologies, it is imperative for scientists to exploit other materials beyond the above materials. Rare-earth oxyborates have caught researchers’ interest, such as the structural disordered style La_3_Ga_5_SiO_14_(LGS), La_3_Ga_5.5_Ta_0.5_O_14_(LGT),La_3_Ga_5.5_Nb_0.5_O_14_(LGN) and the structural ordered style Ca_3_NbGa_3_Si_2_O_14_(CNGS) Ca_3_TaGa_3_Si_2_O_14_(CTGS),Sr_3_TaGa_3_Si_2_O_14_(STGS).

The high-temperature piezoelectric crystal LGS possesses outstanding properties: high resistivity, low dielectric loss and an electromechanical coupling coefficient about 2–3 times higher than that of the quartz crystal.

To promote the properties of this kind of high-temperature piezoelectric crystal, atomic substitution is a general and effective method. For example, Jung et al. prepared the high-temperature piezoelectric crystal La_3_Ta_0.5_Ga_5.5_O_14_ (LTG) with Ga substituted by Al^3+^ [65]. During growth, the substituted Al^3+^ ion altered the distribution coefficient of the modified products with congruent melting in a wide range without destabilizing the melt. The LTGA crystals had dramatically promoted the temperature stability of its piezoelectric properties in the range from *R.T* (room temperature) to 500 °C, indicating its prospects for being integrated in high-temperature applications.

#### 5.1.2. Layer-Structured and Aurivillius Compounds (e.g., Bi_4_Ti_3_O_12_ and Related Materials)

The bismuth layer-structured ferroelectrics (BLSFs) or Aurivillius-type piezoceramics, with their exceptionally high Curie temperature (*T*c) (about 600–900 °C), are the candidates for applications at required temperatures above 400 °C.

(a)Layer-structured (PLS) piezoelectric ceramics

The perovskite layer-structured (PLS) piezoelectric materials have been studied since the 1950s. In 1952, Cook and Jaffe [66] found a piezoelectric ceramic with a A_2_B_2_O_7_ pyroclastic texture that possessed ferroelectricity. In 1955, Jona et al. found Cd_2_Nb_2_O_7_ with a crystalline structure of A_2_B_2_O_7_ [67]. However, not all Cd_2_Nb_2_O_7_ possess ferroelectricity. In 1958, Rowland [68] discovered that Cd_2_Nb_2_O_7_ with a cleavage plane and oxygen octahedral frame have no ferroelectricity. Generally, the PLS piezoelectric ceramics have the structure formula of A_n_B_n_O3_n+2_. The PLS piezoelectric ceramics were stacked by distorted perovskite-type structure layers that are constituted of an oxygen octahedron with apical oxygen atoms BO_6_ and 12 coordination cations. Specifically, n denotes the number of oxygen octahedrons in the perovskite structure layer. As *n* = 4, the formula can be simplified as a A_2_B_2_O_7_-style structure. As *n* = ∞, the formula can be simplified as a ABO_3_-style structure [69].

The perovskite layer-structured (PLS) piezoelectric ceramics mainly contain three kinds of structural styles: the cleavage plane is an orthogonal structure of [010], the cleavage plane is a monoclinal structure of [100], and the surface of the solution is a monoclinic structure of [001] [70].

Taking an orthogonal structure of [010] of Sr_2_Nb_2_O_7_ as an example, the ferroelectric phase at low temperatures is the orthogonal structure with the space group of *Cmc*21. Figure 2a illustrates [71] the crystal structure diagram of the shadow, with the directions cast along the *a* and *c* axes, respectively, where axis *a* is the direction of the oxygen octahedral chain. Meanwhile, the layered distortion of the titanium core can be understood as an octahedron configuration of NbO_6_ with rotations of its small angle, where axis *b* is perpendicular to the direction of the perovskite layer, and axis *c* is situated along the polarization direction. The additional SR–O layers are added between the layers, along the direction of *a*. The additional SR–O is sandwiched between the above two layers and the surface of the solution is a monoclinic structure with the crystalline direction of [010]. The ABO_3_ with a perovskite structure is generally driven by Bi ions (such as BaTiO_3_). The B position is composed of the transition metallic elements with a vacant *d* orbital. Because the *d*_0_ orbital is susceptible to a hybrid of the *2P* orbital with oxygen and the oxygen atoms in the B position may migrate, as a result, the spontaneous polarization occurred [72,73]. In addition, there are small amounts of ferroelectric that were driven by ions in the B position. For example, in BiFeO_3_, Bi has lone pair electrons and is susceptible to hydrating with other orbitals, which may lead to the atoms deviating and the spontaneous polarization phenomenon. From the viewpoint of the crystal structure, although both the perovskite structure and perovskite layer-structured merely contain the oxygen octahedron, the mechanism of spontaneous polarization and structural phase transition of the latter are more complex than that of the former [74].

To investigate the underlying principle of the above phenomenon, theoretical research was also carried out. Jorge et al. [74] simulated the phase transition process of La_2_Ti_2_O_7_ (from ferroelectricity to anti-ferroelectricity) to verify that La_2_Ti_2_O_7_ is the ferroelectricity that was driven by the oxygen octahedrons and the displacement of atoms mainly stemmed from oxygen, while the Ti^4+^ in the B position had no contribution to the displacement of the atom. As shown in Figure 2b, the arrow represents the electric dipole moments generated by the displacement of the oxygen atom. Figure 2b shows the structure of the perovskite layer structure, with total dipole moments of 2p_1_ + 2p_3_ + p_5_. Therefore, the net dipole moment generated by the displacement of the oxygen atom is not zero and exhibits a macroscopic polarization. By contract, with respect to the perovskite structure, the total dipole moment had no contribution to the polarization rate, which can be explained by the fact that the continuous octahedron of oxygen causes its neutralization.

Bruyer et al. studied [75] the electronic structure and ferroelectricity of La_2_Ti_2_O_7_ and Nd_2_Ti_2_O_7_, and the results indicated that their spontaneous polarizations were 7.72 μC/cm^2^ and 7.42 μC/cm^2^. Figure 2c display the electronic density of the state distribution of La_2_Ti_2_O_7_. It clearly can be detected that the bond between Ti and O was not a single ion-bonding, and that the stronger covalent bonding also exists. The hydration of the ferroelectricity phase enhances in orbitals of Ti^4d^ and O^2p^, which lead to the an-isotropicity of electronic density of the state. Sakharov et al. [76] have systematically investigated dielectric constants, piezoelectric coefficients, and thermal expansion factors of PLS piezoelectric ceramics, and found that these parameters were temperature-dependent. Kugaenko et al. [77] also studied thermophysical parameters of langasite (La_3_Ga_5_SiO_14_), langatate (La_3_Ta_0.5_Ga_5.5_O_14_) and catangasite (Ca_3_TaGa_3_Si_2_O_14_) single crystals in a temperature range of 25 to 1000 °C. They found that at elevated temperatures, the heat conductivity anisotropy parameters decreased, and higher χ and λ values were associated with closely packed directions in the crystal.

The typical methods for fabricating PLS piezoelectric materials are solid reaction synthesis, molten salt synthesis and the method of sol-gel [78,79]. The sintering of PLS piezoelectric powders mainly involves solid state reaction (SSR), spark plasma sintering (SPS), hot forging (HF) and so on.

(b)Aurivillius compounds (e.g., Bi_4_Ti_3_O_12_ and related materials)

Bismuth layer-structured ferroelectric (BLSFs) is also named the Aurivillius phase compound, which has an unparalleled Curie temperature, a low dielectric constant, an obvious electromechanical coupling coefficient, a low aging rate and a high resistivity [32]. These merits provide the access to sensor applications for Aurivillius phase ceramics, and they are applied in extreme environments or for nonvolatile ferroelectric random-access memory [80]. The structure can be expressed as (Bi_2_O_2_)^2+^(*A*_m−1_*B*_m_O_3m+1_)^2−^. A represents one valence, di-valence, tri-valence or their combination, constructing the dodecahedron coordination. B represents a transition metallic ion that constructs the octahedron coordination, and *m* represents the number of layers in the perovskite (the value can be selected from 1 to 6) [81]. The common compounds of bismuth layer-structured ferroelectrics (BLSFs) are CaBi_2_Nb_2_O_9_ (CBN) [82], SrBI_2_Ta_2_O_9_ (SBT) [83], Na_0.5_Bi_2.5_Nb_2_O_9_(CN) [84] and Nd_2_Ti_2_O_7_ [85]. *m* = 3 Bi_4_Ti_3_O_12_(BIT) [86] and *m* = 4 CaBi_4_Ti_4_O_15_(CBT) [87]. However, due to its particular 2D structure, the spontaneous polarization is limited in the *a-b* plane [88], which degrades the properties of the bismuth layer-structured piezoelectric crystals: increasing the coercive field causes decreases in the piezeoelectric coefficient, and makes it difficult to obtain a perfect structure by polarization. CaBi_4_Nb_2_O_9_ is a kind of typical bismuth layer-structured ferroelectric ceramic consisting of (Bi_2_O_2_)^2+^(Ca_m-1_Nb_m_O_7_)^2−^ (*m* = 2). The Curie temperature of CBN is drastically high (940 °C), ranking the CBN one of the best high-temperature piezeoelectric materials. It possesses a relative lower piezeoelectric coefficient, with a value of only 5 pC/N, restricting its application in high-temperature areas. Therefore, it is imperative to improve the piezeoelectric properties of bismuth layer-structured ferroelectric materials.

In single crystals, the direction of the spontaneous polarization is determined by the magnitude of the spontaneous polarization. In piezoelectric materials, the direction is determined by the ferroelectric domain state and the maximum magnitude of the spontaneous polarization. In Bi_4_Ti_3_O_12_, the maximum magnitudes of the spontaneous polarization of crystal systems along the orthorhombic [100] and [001] direction are 4 μC/cm^2^ and 50 μC/cm^2^ [89], respectively. The vector of the spontaneous polarization is 50 μC/cm^2^. In orthorhombic crystals, [100] is tilted to [001] by 4.5°. Every component can be calculated by truing the atom [90], the crystal constant and the charge number of the ion. The result is that the crystalline direction [100] titled to [001] by 6.3° is extracted by every polarization component.

The aforementioned Aurivillius-type piezoceramics and bismuth layer-structured ferroelectrics (BLSFs) might be the only available materials which can be operated above 600 °C, taking advantage of their ultrahigh Curie temperature. Apart from their unparalleled high-temperature properties, Aurivillius-type materials are displacement-type ferroelectrics. The spontaneous polarization of bismuth layer-structured compound materials can be ascribed to three aspects: (1) the displacement of the center ion in the octahedronin, (2) the tilting of the octahedronin along the *c*-axis and (3) the rotation of the octahedronin in the *a-b* plane. With respect to bismuth layer-structured compound materials, when *m =* 2*n* and *m =* 2*n +* 1, their spontaneous polarizations are different. When *m =* 2*n,* the spontaneous polarization is along the *a* axis, and when *m =* 2*n* + 1*,* the spontaneous polarization is also along the *a* axis; weak spontaneous polarization also exists along the *c* axis [91].

The spontaneous polarization of bismuth layer-structured compound materials is not the same as a simple perovskite. Newnham et al. assumed in their work that the spontaneous polarization of bismuth layer-structured compound materials originates from the movement of the Ti^4+^ at the Bi position of the octahedronin to edge of the perovskite [92]. Recently, the spontaneous polarization of bismuth layer-structured compound materials can be explained by the fact that the oxygen octahedronin moves along the axis with respect to the Bi ion at the *a* position. The movement can be attributed to the fact that the Bi ion at the *a* position is an unsaturated bond in the primary structure.

The Aurivillius -type piezoceramics also possess the advantages of cost-effectiveness and non-toxicity. Previous studies indicated that most bismuth layer-structured ferroelectric materials possess a 180° domain. However, recently, it was verified that bismuth layer-structured ferroelectric materials have a 180° domain, 90° domain, reverse-phase domain, 180° reverse-phase domain and 90° reverse-phase domain, which were generated by the vanishing of the translation symmetry [93,94,95]. Though, according to space group theory, bismuth layer-structured ferroelectric materials may possess the reverse-phase domain boundary, which does not exist in Bi_4_Ti_3_O_12_. Due to the absence of the reverse-phase domain boundary in the BIT, it is assumed that the interfacial energy of the reverse-phase domain boundary is too high in BIT, causing an unstable interface. Due to the fact that the radius of La^2+^ is bigger than Bi^2+^ and the replacement of La^2+^ for Bi^2+^ may reduce crystal distortions, the interfacial energy of the reverse-phase domain boundary reduces further. As a consequence, the reverse-phase domain boundary takes place.

#### 5.1.3. Perovskites

(a)Universal perovskites materials

Owing to their particular electrical and optical properties, perovskites (HPs) have been “hot spots” with extensive attention paid to it both academically and practically over the past decade. They possess promising prospects in applications for piezoelectric nanowire materials, solar cells and memory devices [96,97,98].

The general formula for perovskites is ABO_3_. The structure of perovskites can be signified as a simple cubic lattice. Every lattice point represents one structural unit. The vertex angle is occupied by a bigger A ion, the center of the lattice is occupied by a smaller B ion and the six oxygen ion takes position in the center of the area. These oxygen ions formed an oxygen octahedron, with the center and gap being taken up by the B and A ions separately. The oxygen octahedron has three fourfold axes, four threefold axes and six bifold axes. The main spontaneous polarization is derived from the movement of the B ion deviating from the center of the octahedron.

In an ABO_3_-type configuration, every ion is distributed in a square model. The main structures of perovskite strongly depend on the tolerance factor parameters (as aforementioned in Section 4.1.2).
(5)$$t=\frac{r_{A}+r_{0}}{\surd 2\left(r_{B}+r_{0}\right)}$$

*t* is referred to as the tolerance factor. Generally, *t* is in a range of 0.77~0.99. To keep the stability of the perovskite’s structure, the distance of the ions changed with coordination numbers, with *t* in a range of 0.77~0.99. For example, in SrTiO_3_, *t* = 1. With respect to the perovskite, when *t* = 0.86 and while in the six-party phase, *t* > 1. The tolerance factor plays an important role for preparing the solid solution. With respect to the high-temperature piezoelectric materials based on perovskites, the characteristics of the electrical domain play an essential role in the performance properties.

In regards to the Pb(Mg_1/2_Nb_2/3_)O_3_-PbTiO_3_, the XRD characterization reveals that the 90° domain has essential effects on the variation of the intensity of the peck, and the 180° domain has no effect on it. Fancher et al. [99] proposed that the XRD pecks of quasi-cubic structures changed from single peaks to double peaks, ascribed to the crystal’ distortions and the rearrangement of the ferroelasticity. With increases in the polarization electrical field, the crystal distortion expanded and the bifurcate became more obvious. Damjanovic et al. [100] studied a single crystal of the PMN-xPT and PN-xPT by in situ measurement. The *DC* electrical field could induce first-order phase transformation. With increasing PT content, the process of phase transformation is R/MA-T, MA-Mc-T and Mc-T, with large quantities of the 180° domain and the 90° domain co-existing in the piezoelectric ceramic. The 90° domain swerve may change the inner stress and induce phase transformation.

The piezoelectricity performance of perovskite piezoelectric materials depends on the composition, crystal structure, grain boundary and electrical domain, as well as the microstructure. Pronin et al. investigated the relationships between the features of the microstructure and the piezoelectric performance of perovskite piezoelectric materials, using perovskite islands in a pyrochlore phase matrix of lead zirconate titanate thin films. The experimental results discovered the effect of the radial inhomogeneity of the surface morphology on the piezo-response of the islands [101]. Doping also can be served an effect method to increase stability. The Pb(Mg_1/3_Nb_2/3_)O_3_(*t* = 0.89) Pb(Zn_1/3_Nb_2/3_)O_3_(*t* = 0.986) can be improved in its stability by replacement of Ba^2+^ with Pb^2+^ [36].

(b)Perovskite materials based on lead-free compounds

Though the perovskite piezoelectric materials based on lead salts show ultrahigh piezoelectric responses, the toxicity issue poses severe challenges for their broad application. Therefore, it is urgent to exploit for lead-free piezoelectric materials [35,102,103,104,105,106,107].

With respect to the mechanism of lead-free perovskite piezoelectric materials, no unanimous consensus has been arrived at. Among the myriad of high-temperature piezoelectric materials, KNN can also be recognized as a star material. Ka_0.5_Na_0.5_NbO_3_ [108] is composed of anti-ferroelectrics (NaNbO_3_) and ferroelectrics (KaNbO_3_). The KNN ceramic also possesses a high *T*c and high *k*_p_. Until now, the KNN ceramic is one of the high-temperature piezoelectric materials that simultaneously possess comparatively good piezoelectric properties and Curie temperatures. However, the drawbacks regarding the polymorphic phase transition and unfeasible processing procedures have significantly restricted its development as a lead-free high-temperature piezoelectric material.

### 5.2. The Hybrid High-Temperature Piezoelectric Materials

It is widely acknowledged that to boost the piezoelectric performance of pure piezoelectric materials, incorporating other components to construct hybrid systems demonstrates a powerful approach to make up for the shortcomings of each component and to exploit synergy effects [109,110,111,112,113,114]. Baur et al. [109] enhanced the piezoelectric performance of PVDF with carbon fluoropolymer nanocomposites. This is also the case with high-temperature piezoelectric materials, though reports on hybrid high-temperature piezoelectric materials are really rare. The incorporation of foreign components may effectively elevate the intrinsic temperature of the materials and even render the piezoelectric materials pristine, and when applied in *R. T.*, it can be used at high temperatures. As an example, Yang et al. [115] designed a flexible PVDF-based nanocomposite, with the added components considerably promoting the inherent usage temperature of the PVDF, showing a stable piezoelectricity covering a wide temperature range from 10 to 100 °C. The advanced performance is mainly attributed to the strong multi-interface effect. Meanwhile, the piezoelectric coefficient *d*_33_ reached a peak of 40 pC/N and showed excellent retention with increasing temperatures up to 100 °C.

Jian et al. [116] developed a method for synthesizing PbTiO_3_ with a 3D multilevel flower-like structure (PTFs) and polyimide (PI)/PTFs composites to prepare piezoelectric generators (PENGs) with outstanding energy harvesting performances at high temperatures. The PENGs made from the PI/PTFs composite showed excellent output performances of *I*_SC_ ~ 103 μA, *V*_OC_ ~ 140 V and P ~ 2128 μW. Particularly, higher degrees of electric power can be continuously generated by the PENG at a wide range of temperatures, from room temperature to 300 °C. Excellent durability up to 5000 cycles can also be achieved for the PENG within the temperature range.

As aforementioned in Section 4.1.3, Guo et al. [40] fabricated high-temperature piezoelectric materials of BiFeO_3_–BaTiO_3_–NdCoO_3_ with multiferroic characteristics, and uncovered that the introduction of NdCoO_3_ into BiFeO_3_–BaTiO_3_ considerably promoted its piezoelectricity and multiferroicity, and the magnetoelectric effect of the materials.

As aforementioned in Section 4.2.2, via incorporating the content of the second unit to form hybrid nanocomposites, phase structures also change, giving rise to the morphotropic phase boundary (MPB) or polymorphic phase transition [49]. Chen et al. prepared the piezoelectric ceramics (1–x) Sr_2_Nb_2_O_7_(Na_0.5_Bi_0.5_)TiO_3_(x = 0~0.05) through the traditional sintering method by incorporating perovskite Na_0.5_Bi_0.5_TiO_3_ into the Sr_2_Nb_2_O_7._ The incorporation of Na_0.5_Bi_0.5_TiO_3_ had remarkably improved the properties of Sr_2_Nb_2_O_7_. The obtained non-textured piezoelectric ceramics based on Sr_2_Nb_2_O_7_ had excellent piezoelectric performance with *d*_33_: ~1.0 pC/N and good thermal stability. The de-polarization temperature was above 1200 °C [117].

The newly merged 2D materials have injected fresh blood into the family of high-temperature piezoelectrics materials. Mohanta et al. have [118] explored a novel high-temperature piezoelectric material based on the 2D material BP-gallium nitride, and constituted a stable 2D van der Waals heterojunction. The remarkably high piezoelectric coefficient was (*d*_33max_ ≈ 40 pm V^−1^), which originated from the large difference in atomic charges between the GaN and BP monolayers and the out-of-plane inversion asymmetry, This value was comparatively higher than the out-of-plane piezoelectric coefficient of the multilayered Janus transition metal dichalcogenide MXY (*d*_33max_ = 10.57 pm V^−1^) in the previous work. This state-of-the-art BP/GaN hetero-bilayer could open new opportunities for next-generation devices.

## 6. The Design of the High-Temperature Piezoelectric Materials

### 6.1. How Do the Materials in Section 5 Meet the Limiting Factors Identified in Section 4?

The typical high-temperature piezoelectric materials are introduced above. Practically, according to their own characteristics, the targets and the anticipated effects of the projected properties of the materials are expected to be taken into comprehensive consideration.

(a)The investigation of phase characteristics

The phase characteristics can be recognized as one of the decisive factors on the utmost properties of high-temperature piezoelectric materials. Due to the narrow window for preparing BiFeO_3_, the products may be mixed up with a second phase. During the mixing procedure, Fe^3+^ is susceptible to transiting to Fe^2+^ and giving rise to the BiFeO_3_, accompanied by the production of a large quantity of oxygen vacancies. As a consequence, the saturated ferroelectric hysteresis loop is undetected. To improve the properties of BiFeO_3_, element doping is applied (e.g., La [119], Tb [120], Mn [121], Ti [122], Dy [123]). To obtain a stable perovskite structure, it is reasonable to construct a solid solution system with another ABO_3_-style ferroelectric (e.g., BiFeO_3_, PbTiO_3_, PbZrO_3_, Pb(Fe_0.5_Nb_0.5_)O_3_ [124]). The incorporation of BiFeO_3_ into the perovskite structure-based piezoelectric materials can be stabilized and limited in its foreign phase, with its resistivity improved as well [125].

The piezoelectric materials with an Aurivilius-type structure also experience phase transition. Steiner et al. have [126] investigated the piezoelectric performance of bismuth titanate niobate Bi_7_Ti_4_NbO_21_, which was closely related to the Bi_3_TiNbO_9_ clusters during the phase transition process.

(b)The polarization

The polarization powders technology is utilized to analyze the crystal structure and dielectric coefficient of the tetragonal phase that is located far from the composition points, as well as the tetragonal phase and monoclinic phases that are located near the composition points. Moreover, this technology is employed for truing the MPB and the rhombic phase before polarization and after polarization [127]. Through this technology, it can be observed that the rhombic phase after being polarized virtually contains two structures, *Cm* and *P*4*mm*, explained by the fact that the metastable structure of the second phase expanded during the polarization process. It can also be detected that the dielectric coefficient increased at the MPB composition points and near the MPB composition points after being polarized. At the lower composition points, the dielectric coefficient after being polarized is lower than it is before being polarized. The decrease in the dielectric coefficient was attributed to its alignment along the orientation of the polarization. By contrast, the increase in the dielectric coefficient was derived from the phase transition of ferroelectric materials. In addition, with respect to the BNT relaxation ferroelectrics and PMN relaxation ferroelectrics, it could be deduced that the polarization electrical field had effects on the crystal structure to the same extent [128].

(c)Structural modification and process optimization

Currently, a great many scientists endeavor to improve the piezoelectric properties by component optimization, phase structural engineering and process improvement. Particularly, the structural engineering is also important for promoting the piezoelectric performance of piezoelectric materials for high-temperature applications. For instance, Wang et al. have [129] presented a lithium niobate single crystal possessing the largest effective piezoelectric and electromechanical coupling factors by an optimum crystal cut (XZt/28°), and indicated that the structural modifications have pronounced influence on the macroscopic electrical properties and, furthermore, lead to their high Curie temperature and piezoelectric activity.

The high-temperature piezoelectric ceramic Pb_1-x_(Yb_1/2_Nb_1/2_)_0.515_Ti_0.485_O_3_-x{(2SrO + BaO)/3} [PYNT–Sb_x_] can be altered by atomic Sb substitution at the rhombohedral side near the morphotropic phase boundary to increase the tetragonality. The Sb-doped product changed the function relationship between the temperature and the relative permittivity. The piezoelectric coefficient dramatically improved to 610 pm V^−1^ and the maximum dielectric constant soared to 25,000 [130].

(d)The investigation of doping

In Section 4.1 of this review, we mentioned that the Curie temperature could be evaluated by doping. Doping is a general method to increase the Curie temperature and the high-temperature stability of high-temperature piezoelectric materials. The most widely used dopants are salts. Scientists have also made efforts to gain deeper insights into the underlying mechanisms of doping. Lin et al. have promoted the piezeoelectric coefficient *d*_33_ from 93 pC/N to 208 pC/N by doping the Li and Ta elements [131]. Previous studies reckoned that the improvement of the piezeoelectric properties was ascribed to the effect of doping of the constructed MPB phase boundary (just like the boundary in PZT). However, the strengthening mechanism of piezeoelectric ceramic of (K, Na)NbO_3_ was polymorphic phase transition (PPT) rather than the typical MPB phase boundary. It is acknowledged that the MPB possessed excellent thermal stability, which was closely associated with the components of the piezeoelectric ceramic. By contrast, the PPT mechanism is strongly temperature-dependent. That is, the promoting of piezeoelectric properties of (K, Na) NbO_3_ can be explained by the fact that the transformation from the orthogonality phase to the tetragonal phase is controlled in room temperature rather than 200 °C. Nevertheless, the PPT effect has a negative effect on the thermal stability and is harmful for the realization of commercialization.

### 6.2. The Analysis of Performance

(a)The inherent structure (comparison of nonferroelectric piezoelectric crystals and ferroelectric crystals)

The nonferroelectric piezoelectric crystals (e.g., quartz crystals, AlN crystals) also show high Curie temperatures, yet, their piezoelectric properties are unsatisfied. This can be ascribed to the absence of spontaneous polarization and ferroelectric-related electroporation within these crystals, which can be explained by the fact that the piezoelectricity of these crystals stems from the asymmetry of the crystal structure, not their ferroelectricity. On the contrary, ferroelectric materials can produce a larger piezoelectric response depending on the orientation or flip of the ferroelectric ceramics under the electric field, motivating many researchers to attempt to fabricate the single crystal of ferroelectric ceramics with a high Curie temperature [132].

(b)The relationship between Curie temperature and the tolerance factor

In regards to some perovskite-based high-temperature piezoelectric ceramics, the relationship between the Curie temperature of the materials and the tolerance factor is almost linear. As shown in Figure 3, the smaller the tolerance factor, the higher the Curie temperature, with the Curie temperature of the material smallest as the tolerance factor moves closer to 1 [37]. Until now, there is no exact interpretation for the above relationship. According to the definition of the tolerance factor, when the crystal adopts the cubic phase (paraelectric phase), the tolerance factor becomes 1. The smaller the tolerance factor, the greater the structural distortion of the material with regard to the deviation of the crystal structure from the cubic phase, which is responsible for the increase in the Curie temperature with increasing structural distortions. This may explain why the smaller the tolerance factor, the higher the Curie temperature.

The tolerance factor heavily depends on the elementary composition of the materials. As the components of ABO_3_ are Pb (Mg, W) O_3_, its tolerance factor is 0.993 and its Curie temperature is 60 °C. For BiYbO_3_, the tolerance factor decreases to 0.857 and the Curie temperature increases to 613 °C.

(c)The relationship between the Curie temperature and the piezoelectric coefficient

Besides the tolerance factor, the Curie temperature is also associated with the piezoelectric coefficient *d*_33_ of the piezoelectric materials. Generally, the higher the Curie temperature is, the lower the *d*_33_ of the materials turns out to be. The piezoelectric coefficient of common piezoelectric ceramics and the variation of the Curie temperature relative to its components are illustrated in Figure 4. The piezoelectric coefficient of common PZT piezoelectric ceramics is higher than 300 pC/N, but the Curie temperature is generally not more than 400 °C.

Figure 4 displays the variation of the Curie temperature and the piezoelectric coefficient in general piezoelectric ceramics with varied components [133]. The piezoelectric coefficient of common PZT piezoelectric ceramics is higher than 300 pC/N, with a relatively lower Curie temperature (no more than 400 °C). Both the perovskite-like layer-structured piezoelectric ceramics and the bismuth layer-structured oxides piezoelectric ceramics possess high Curie temperatures, yet their piezoelectric coefficients are lower than 30 pC/N. There are still rare mechanisms for elucidating the contradictory relationship between piezoelectivity and Curie temperature. The quantitative relationships between them have also been in the process of being researched. So, phenomenological theory can be adopted to explain it intuitively. That is, piezoelectivity is associated with the intensity of the spontaneous polarization inside the materials. A greater spontaneous polarization intensity brings a greater piezoelectivity, leading to a decrease in thermal stability. The improving of the Curie temperature and piezoelectivity simultaneously are long-sought goals which need a systematic theoretical system to elucidate their internal mechanisms.

(d)Depolarization

Depolarization is an undesirable but important phenomenon which significantly degrades piezoelectric materials. Unfortunately, there is no effective and accurate measurement method to evaluate the degree of depolarization of piezoelectric materials. An in situ quasi-static high-temperature Berlincourt test is introduced to measure the piezoelectric constants and extract the relationships between the piezoelectric coefficients and the temperature. This shed light on the studying of the depolarization phenomenon [42,134].

### 6.3. Research Technique

To quantitatively optimize the parameters for Curie temperature, the tolerance factor and piezoelectivity, the phase field technique [135] can serve as a powerful tool to determine the optimal conditions experimentally. For example, this technique can give the most-rational designs by means of investigating the interaction mechanism of grain size [136] and oxygen activity [137] on the microstructures and properties of BaTiO_3_ ceramics. Narita et al. reported [138] a phase field simulation framework to study the effects of oxygen vacancy densities and grain size on the poling of the pristine BaTiO_3_ polycrystals, as well as on the piezoelectric coefficient and the permittivity of the poled BaTiO_3_ polycrystals. Additionally, phase field simulation [139] also can be introduced to study the intrinsic relationships between the grain size on the domain structures and the electromechanical properties of ferroelectric polycrystals. Apart from phase field simulations, finite element analysis is also an effective approach to deeply and comprehensively analyze porous sandwich structures, opening a new avenue for designing and optimizing the structure and procedure of the piezoelectric materials.

## 7. Applications of Piezoelectric Nanomaterial at High Temperature

### 7.1. General Applications of Piezoelectric Nanomaterial at High Temperature

#### 7.1.1. High-Temperature Piezoelectric Drivers

High-temperature piezoelectric drivers are one of the important applications of high-temperature piezoelectric materials. Its large load, high precision, high efficiency, fast response, lack of an electromagnetic interference and other advantages spur widespread attention. High-temperature piezoelectric drivers are generally applied in high-temperature piezoelectric valves for fuel injection. In order to improve the combustion efficiency of diesel engines, multilayer pressure electrical actuators have been widely used to replace solenoid valves, which can effectively improve fuel combustion efficiency and reduce CO_2_ and NOx emissions. However, the temperature of the injection chamber is over 150 °C. After being used at this temperature for a long time, traditional PZT piezoelectric ceramic materials will generate aging and degrading problems. In the field of national defense, high-temperature multilayer piezoelectric drivers directly control the valve-to-pulse width of the lateral bypass of tail nozzles. Control of direct lateral forces is an important means for controlling missiles. Recently, owing to its higher Curie temperature and higher piezoelectric coefficient *d*_33_, the BS–PT-based high-temperature piezoelectric ceramics have been expected to replace PZT piezoelectric ceramics and overcome the existing problem of high-temperature aging. The left panel of Figure 5a shows the multilayer piezoelectric drivers made of BS–PT ceramics. The strain and displacement fractions of a multilayer BS–PT piezoelectric actuator at the temperature point 200 °C were obtained under a 7.5 kV/cm electric field, reaching up to 0.115% and 11.5 µm at 25–200 °C, respectively [140]. The right panel of Figure 5a shows a schematic diagram of a high-temperature piezoelectric multilayer drive applicator in the injection valve of a diesel engine.

#### 7.1.2. High-Temperature Piezoelectric Transducer

Piezoelectric actuators convert mechanical energy (e.g., mechanical vibrations, displacement) to electrical energy by the inverse piezoelectric effect. Wu et al. [141] prepared a piezoelectric ceramic material based on BS–PT with piezoelectric vibrations comprised of a *d*_31_-mode cantilever beam structure used in high-temperature energy reclaimers, and overcame the drawbacks of the conventional cantilever PZT piezoelectric (vibration energy recoverer recycler, due to the relatively low Curie temperature, as well as the usage of epoxy trees) which seriously restrict its application in high-temperature environments. The upper panel of Figure 5b shows the schematic diagram of the junction of a high-temperature energy collector with a *D*_31_ mode. The bottom panel of Figure 5b illustrates a schematic diagram of a high-temperature test device; the device adopts a machinery clamping structure that replaces the epoxy resin bonding, avoiding degradations of the performance of the device’s parts due to epoxy resin failure.

#### 7.1.3. High-Temperature Acoustic Emission Sensor

Acoustic emission sensors work in the broadband response frequency range. Though ferroelectric material-based transmitting acoustic emission sensors are widely used, they are limited by their material properties that are incompatible with applications at 500 °C and above. Johnson et al. used a non-iron piezoelectric acoustic emission sensor, fabricated by YCOB single crystal material [142,143]. The structure of the device is shown in the top panel of Figure 5c. The temperature ranges from room temperature to 1000 °C, and the setup device is shown in the bottom panel of Figure 5c.

#### 7.1.4. High-Temperature Piezoelectric Vibration Sensor

The engine is the heart of the aerospace vehicle, and the real-time and dynamic monitoring of the motion inside the engine is directly related to the aircraft. In order to accurately test the motion state of various components in the engine under a high-temperature environment, the sensor is expected to be placed directly on the engine’s surface or blade at temperatures above 1000 °C. Because of its stable piezoelectricity and electromechanical coupling properties when performing at high-temperatures, the YCOB crystal has received widespread attention. Kim et al. [144] designed and prepared a type shear high-temperature accelerometer based on the YCOB crystal for temperatures above 1000 °C. The structure of the vibration sensor at the above ambient temperature is shown in the upper panel of Figure 5d, and the high-temperature environment test equipment is shown in the bottom panel of Figure 5d.

#### 7.1.5. The Acoustic Wave (SW) Devices and Surficial Acoustic Wave (SAW) Devices

High-temperature piezoelectricity has promising prospects in acoustic wave (SW) devices and surface acoustic wave (SAW) devices. Because of the low transmission velocity of surface acoustic wave (SAW), the size of the device may decrease significantly, which may greatly motivate the progress of a miniaturized device. Furthermore, the composite material for acoustic wave (SW) devices requires the crystal to have no phase transitions in the temperature range of 1300–1500 °C. For example, ReCOB, a hot-topic material for high-temperature applications, has no phase transition until melting temperatures and possesses a high electromechanical coupling coefficient and a high piezoelectric coefficient. Besides, it also possesses high electromechanical temperature stability (at 1000 °C the piezoelectric coefficient is almost unchanged) and unprecedented resistivity. Particularly, the crystal is grown by the *Czochralski* method, which is responsible for it being a high-quality large-area crystal, demonstrating its potential for new-generation piezoelectrics for high-temperature applications. SAW sensors can test the temperature by wireless transmission, and are placed inside the engine. Aubert et al. [145] studied AlN/ sapphire and indium forked fingers and tested the long-term working stability of the device. The structure of the electrode used at a high temperature (1050 °C) is shown in the upper panel of Figure 5e. As shown in the bottom panel of Figure 5e, though, at 1050 °C the AlN layer underwent severe oxidation, but showed good stability below 1000 °C.

Kuznetsova et al. studied the temperature coefficient of delay for lowest-order plate wave modes, S_0_, A_0_, and SH_0_, in the LiNbO_3_ crystal. TCD is demonstrated to be a weak function of the normalized plate thickness h/λ (h = plate thickness, λ = acoustic wavelength) in almost all modes and in all orientations, except for the A_0_ mode in X-Z and the YZ plates. The results showed that plate waves can provide higher *K*_2_ and lower TCD than is possible with SAWs, which conforms well with theory [146].

#### 7.1.6. The Frequency-Controlling Devices

The frequency-controlling devices of some equipment for aeronautics and astronautics or other precise instruments are generally operated at high temperatures, which puts forward higher requirements for piezoelectric nanocomposites, and they inevitably play essential roles in this area. For instance, Karaki et al. monitored the electric elastic constant of CNGS crystals by the resonance method and measured the temperature coefficient of the elastic constant, demonstrating its promising application for frequency-controlled devices [147].

#### 7.1.7. The Vibrating Beam Resonator

Douchet designed a vibrating beam resonator with the crystals possessing a 32-point group, and revealed that the crystals with length-stretching vibration mode possessed the zero-frequency cut symbol [148].

#### 7.1.8. The Sensors

Fritze et al. systematically investigated the BAW resonator of LGS crystals in high temperatures, and designed the LGS gas sensor on the basis of this [149].

Tortissier et al. studied the sensor of LGS crystals employed in detecting chemical components in harsh environments. They also evaluated the frequency variation of the retardation line [150].

Anisimkina et al. presented temperature sensor prototypes based on a surface acoustic wave with high-sensitivity selectivity, which can be operated independently of the rest of the physical parameters of liquids. The ultrahigh selectivity (0.03–0.005 °C) is realized by preventing physical contact between a tested sample and a surface acoustic wave. With the aim of enhancing the sensitivity, they increase the wave path from a transmitter to a receiver, accompanied by a fifth wave harmonic [151].

Besides the above sensors, gas sensors are also important. During high-temperature combustion, a piezoelectric gas sensor is adopted for closely monitoring ambient gas and pressure. The microbalance working mode adsorbs and compresses gas molecules through thin films. The resonant frequency of an electrical element varies with its mass [152,153]. The upper panel of Figure 5f illustrates the work principle of the gas sensor and the bottom panel of Figure 5f shows the monitoring result of the oxygen content in a hydrogen/argon environment by an LGS gas sensor. It can be seen that the device can achieve gas monitoring in a range from room temperature to 600 °C. Takeda et al. [154] presented a pressure sensor based on calcium aluminate silicate (CAS) crystals and the performance test under dynamic stress loading.

#### 7.1.9. The Optoelectronic Devices

Besides the aforementioned applications, high-temperature piezoelectrics nanocomposites can also be adopted for structuring optoelectronic devices. It is worth mentioning that the crystal structure of AgGaGeS_4_–AgGaGe_3_Se_8_ possesses the particular optoelectronic characteristics, showing great promise for optoelectronic applications. For instance, Naggar et al. [155] designed a novel and different laser-powered optoelectronic device based on a chalcogenide semiconductor. The optoelectronic features of the high-temperature piezoelectric materials originated from their intrinsic defects and the electron-phonon anharmonicities within their crystal lattice. SEM and EPMA revealed that AgGaGeS_4_ and AgGaGe_3_Se_8_ single crystals possessed high-quality and homogenous characteristics. Particularly, the well-defined Ag-Ga-Ge-(S,Se) crystals and solid solutions had the advantages of high birefringence and high non-linear optical efficiencies, demonstrating their great prospects in coherent laser frequency conversion in the near-infrared spectral range.

### 7.2. Physical Properties for Practical Application

The unsatisfying speed of development of high-temperature piezoelectric materials has seriously impeded the use of piezoelectric devices in extremity environments. The high-temperature applications not only propose advanced requirements for its material properties (the half-Curie temperature point is supposed to be higher than the operational temperature, and the piezoelectric constants *d*_33_,*d*_31_ are required to be stable across the whole temperature range, from room temperature to operation temperature), *d*_33_ also requires a high thermal stability [156] at the operating temperature. Secondly, during the process, it is supposed to avoid adhering to resin and other organic substances. In regard to some special materials, it is supposed to guarantee that all materials in the whole structure possess the ability for withstanding the test of a high-temperature environment [157,158]. For a transducer, the most important factors are the variation of its piezoelectric constants, electromechanical coupling coefficients and dielectric constants with regard to temperature and vibration. Specifically, as the temperature moves above 500 °C, the requirements for ultrasonic transducers are higher. Therefore, researchers resorted to the materials possessing high Curie temperatures (YCOB, LN crystals) [143,159]. Thirdly, the thermal fatigue performance is also required to lay emphasis on high-temperature environments, for the depolarization effect is more severe when the temperature and vibration are imposed simultaneously. The depolarization of the piezoelectric constant *d*_31_ is easier than *d*_33_.

For high-temperature sensors, temperature stability is required to be given the highest priority. It is urgent for high-temperature accelerometer materials to maintain high-temperature stability, high resistivity, narrow bandwidth, low mechanical loss, and a high electromechanical coupling coefficient [160]. The stability is required to be given priority for the materials used in high-temperature devices (e.g., SAW) [145], while high resistivity, the temperature expansion coefficient, the compliance coefficient and mechanical loss are also required to receive emphasis. So, the high-temperature crystal YCOB has become the most favored material [161,162] for these high-temperature sensors.

## 8. Concluding Remarks

Due to the rapid progress that has been made in material science, electronics and energy science, high-temperature piezoelectrics will be given considerable attention to fulfill the demands of crucial devices in aeroengines, nuclear reactors and so on. Great progress has been made to improve the properties of piezoelectric materials that are applied in elevated temperatures. Through systematically introducing the fundamental knowledge about the piezoelectric mechanism, piezoelectronics and piezophotoelectronics, and describing the classification, applications, parameters and typical instances of high-temperature piezoelectric materials, this review offers relatively comprehensive information for scientists or researchers to design and explore high-temperature piezoelectric materials. However, there are still many challenges for the development of high-temperature piezoelectric materials. First, hitherto, there is rarely a principle that explains the relationships between the microstructure and properties. Second, it is un-accessible to achieve the synergistic interactions among the multiple factors with the optimum points. Third, it is imperative to exploit the comprehensive properties combining ultrahigh de-polarization temperatures, piezoelectricity, a low damping and an outstanding mechanical and flexibility for meeting the new demands of the area of sophisticated equipment.

## Figures and Tables

**Figure 1 nanomaterials-12-01171-f001:**
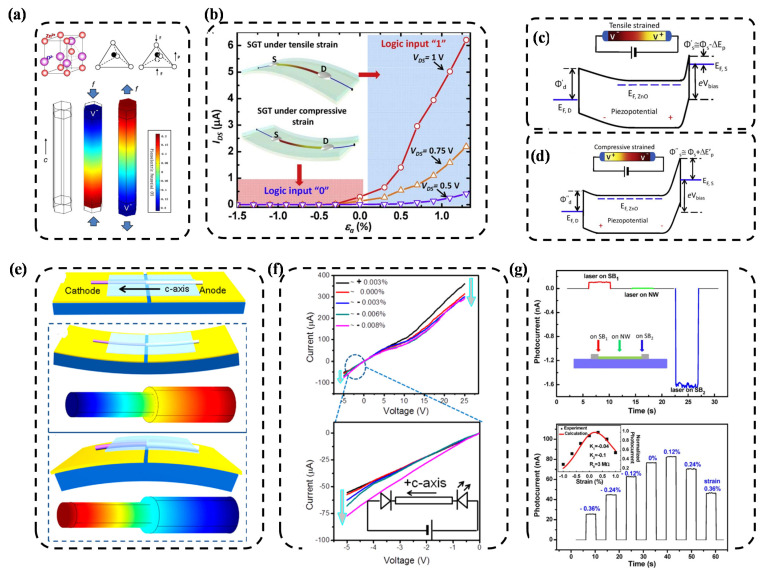
Fundamentals of the piezoelectric effect. (**a**) Crystalline model of the wurtzite-structured ZnO (upper panel); numerical simulation of the piezoelectric potential distribution in a ZnO nanowire under axial strain. The growth direction of the nanowire is the c-axis. The dimensions of the nanowire are L = 600 nm and a = 25 nm; the external force is fy = 80 nN (bottom panel). Reproduced with permission from Ref. [10]. Copyright 2009 American Institute of Physics. (**b**) Transfer characteristics of the current strain (*I*_DS_-*ε*) for a ZnO SGT device, with the strain sweeping from ε= −0.53% to 1.31% at a step of 0.2%, where the *V*_DS_ bias values were 1 V, 0.75 V and 0.5 V, respectively. (**c**) Under tensile strain, the SBH at the source side is reduced from φ_s_ to φ_s_′ ≈ φ_s_ − ΔE_P_. (**d**) Under compressive strain, the SBH at the source side is raised from φ_s_ to φ_s_″ ≈φ_s_ + ΔE_P_′. (**e**) Schematic of the device (top panel), stimulation of piezopotential at compressive strain state, (middle panel) and at tensile strain state (bottom panel). The color gradient represents the distribution of piezopotential, with red for positive and blue for negative potentials. (**f**) Upper panel: piezotronic performances of the device under various strain levels. (**e**) Bottom panel: magnification of the characteristics for the negative voltage range. Reproduced with permission from Ref. [13]. Copyright 2012 American Chemical Society. (**g**) Maximizing the photocell output by the piezophototronic effect. Upper panel: The output current, as the laser spot, was focused at different positions of the wire. The inset shows the sketched picture to indicate the related illuminating position of the laser on the device. Bottom panel: Output current responses to the strain applied on a device. The inset is the simulated result based on an equivalent circuit model. Reproduced with permission from Ref. [16]. Copyright 2013 American Chemical Society.

**Figure 2 nanomaterials-12-01171-f002:**
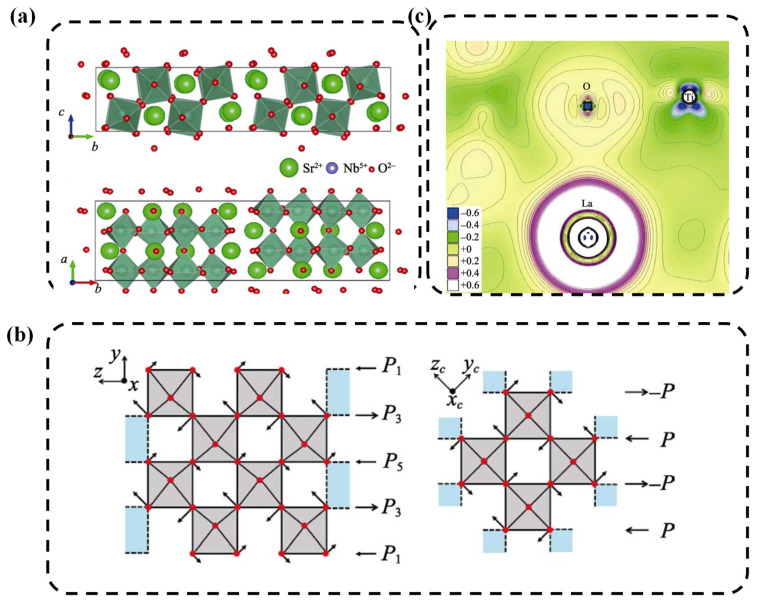
The perovskite layer-structured (PLS) piezoelectric ceramics. (**a**) Crystal structure of Sr_2_Nb_2_O_7_ with visual angle along *a* (top panel) and along *c* (bottom panel). Reproduced with permission from Ref. [69]. Copyright 1975 Science Direct. (**b**) Left panel: schematic diagram of the largest atomic displacements associated with the strongest instability mode obtained for the *Cmcm* phase of La_2_Ti_2_O_7_; Bottom panel: schematic diagram of a typical anti-ferrodistortive mode occurring in an ideal perovskite structure of BaTiO_3_. The arrows on the side represent the electric dipoles associated with the displacement of oxygens in a different y plane. Reproduced with permission from Ref. [74]. Copyright 2011 ISOLED. (**c**) Extracted valence-electron charge density for the monoclinic *P*_21_ La_2_Ti_2_O_7_. Contour lines differ by 0.05 e Å−3(1 Å = 0.1 nm). Reproduced with permission from Ref. [75] Copyright: 2010 American Institute of Physics.

**Figure 3 nanomaterials-12-01171-f003:**
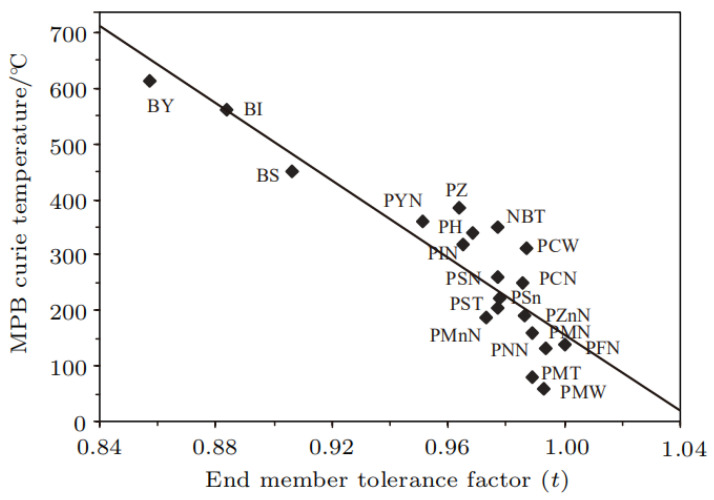
The relationship of tolerance factor *t* and Curie temperature of MPB components in (1-x)ABO_3_-xPbTiO_3_ (ABO_3_ components) solid solution. Reproduced with permission from Ref. [37]. Copyright 2001 American Chemical Society.

**Figure 4 nanomaterials-12-01171-f004:**
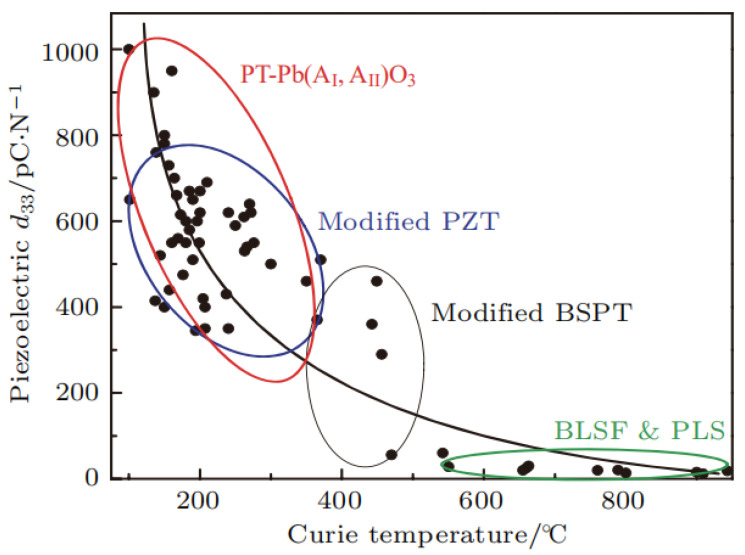
Piezoelectric coefficients and Curie temperature of common piezoelectric ceramics relative to components. Reproduced with permission from Ref [133]. Copyright 2010 American Chemical Society.

**Figure 5 nanomaterials-12-01171-f005:**
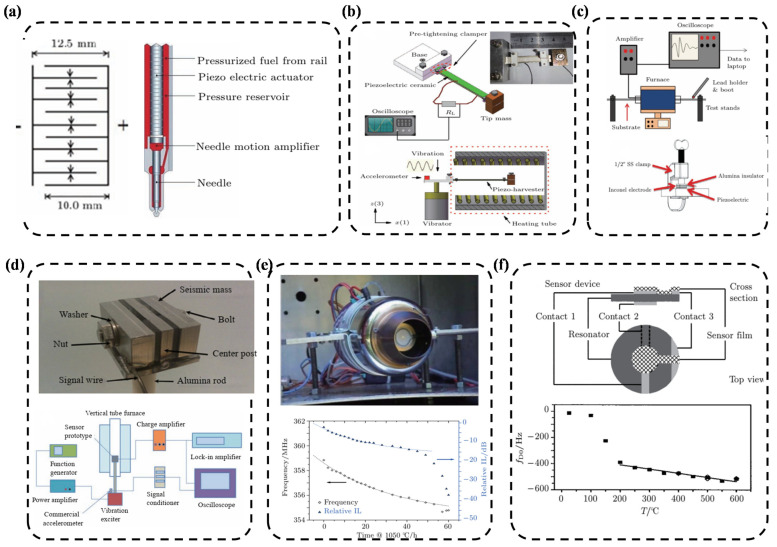
Applications for piezoelectric nanomaterial at high temperatures (**a**) left panel: High-temperature multilayer drivers can be safely used at temperatures up to 250 °C; right panel: lectro-pressure injection valve. Reproduced with permission from Ref. [140]. Copyright 2012 Elsevier Science Direct. (**b**) Schematic diagram of a high-temperature energy recovery device in *D*_31_ mode (upper panel). Schematic diagram of temperature test device (bottom panel). Reproduced with permission from Ref. [141]. Copyright 2016 Wiley. (**c**) YCOB-based high-temperature acoustic emission sensor samples (upper panel) and the HSU-Nielsen acoustic emission test setup (bottom panel).Reproduced with permission from Refs. [147,148] Copyright: 2018 American Institute of Physics. (**d**) YCOB-crystal-based high-temperature piezoelectric vibration sensor (upper panel) and high-temperature sensor experimental test equipment (bottom panel). Reproduced with permission from Ref. [144] Copyright: 2012, Elsevier Science Direct. (**e**) Schematic diagram of Al/ Sapphire SAW device at high temperatures (upper panel) and test results at 1050 °C, with different frequencies (bottom panel). Reproduced with permission from Ref. [150] Copyright: 2001 American Institute of Physics. (**f**) Working principle of gas sensor (upper panel). Reproduced with permission from Ref. [152] Copyright: 2001, Elsevier Science Direct. Bottom panel: monitoring of oxygen content in hydrogen/argon environment by LGS gas sensor. Reproduced with permission from Ref. [153] Copyright: 2001, IEEE.

## Data Availability

The study did not report any data.

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
