# Peer review of "Piezoelectric Materials: Properties, Advancements, and Design Strategies for High-Temperature Applications"

_nanomaterials, 2022, doi:10.3390/nano12071171_

Round 1

Reviewer 1 Report

Meng and coworkers, report a very detailed, rich of content and elucidative review paper entitled “Nanomaterials for High Temperature Piezoelectric Applications”.

In order to sound properly, few major changes are requested:

1) The title does not sound appropriate: only 1 page (section 7) out of 33 pages of the review is really addressing the high temperature applications. The title must be representative of the manuscript, if not it is misleading for the readers. In addition, the materials are piezoelectric, not the applications. I suggest rethinking the title to give a better balanced summary of the review’s story. For example: “Piezoelectric nanomaterials: properties, design factors and advancements for high temperature applications”… or something similar.

2) Even though the review is highly comprehensive of concepts to guide the readers to the understanding of high temperature piezoelectric materials, the final part on the applications is insufficiently addressed and should have a higher weight in the manuscript. Maybe the authors can bring some specific example that they think is innovative and insist on future perspectives. It is not clear (as stated in the abstract) why the health monitoring applications should be considered “high temperature”.

3) The review has only 4 figures. I suggest to draw some schematics or insert figures (e.g., from open-access sources, or others that give the copyright permission for reuse), in particular in the applications. This would further remark the practical interest in these materials and clarify where they can be applied.

Minor comments:

Place “eq. 1”, “eq. 2” and so on in line with equations.

References are not superscripted in this journal guidelines.

Line 80: Equation has a grey background. Should the authors use the equation editor embedded in Microsoft word template?

Figure 1 contains name and number fonts which are too small to be read. Please make the figure bigger; figure caption may stay in the next page.

Line 254: Equation is in bold. Please make the normal font type.

Line 260: “4.1.1. The Important of Curie Temperature”, change it with ““4.1.1. The Importance of Curie Temperature”

Lines 758-762: “b) Perovskites materials based on lead-free compound” and similar. Perovskite is not written plural, as it behaves like and adjective of the subsequent name (materials); please change into “b) Perovskite materials based on lead-free compound”. Edit similar typos in other sentences.

Figures 3 and 4 must be bigger to be better appreciated.

Author Response

To the Reviewer:

Thank you for your reviewing on our manuscript “Nanomaterials for High Temperature Piezoelectric Applications” co-authored by Yanfang Meng et al. with provide enthusiastic, positive comments. We have revised the manuscript based on the your comments. We thank your comments/ suggestions and your efforts, which have helped to improve the overall quality of the manuscript.

The following are our replies (marked in blue) to the reviewers and modification to the manuscript (marked in red). We hope the revised manuscript meets the high standard of Nanomaterials.

We looked forward to hearing from you.

Yours sincerely,

Yangfang Meng

State Key Laboratory of Advanced Optical Communications System and Networks, School of Electronics Engineering and Computer Science, Peking University,Beijing 100871, China.  Center for Flexible Electronics Technology, Tsinghua University, Beijing, 100084, China.

Tel: 86-(10)62784622 

Reviewer 2 Report

The Review presented in the manuscript is very interesting and useful for scientists working in fields of acoustoelectronic devices developement that are have possibility to work under high temperature. But I have some notes:

  1. Title: review devoted mainly not nanomaterials but composite materials. So it is better to use in the Title and in the manuscript (for example lines 58, 62, 262 and so far) instead “nanomaterials” the term “composite materials” or “nanocomposite materials” where it is appropriate (item 5.2).
  2. In Introduction line 40 papers [3,4] were published in 1991 and 1999, correspondingly. It will be good add more fresh papers devoted to discussed topic. It is possible to add

- Caliendo C., Latino P.M. Characterization of Pt/AlN/Pt-based structures for high temperature, microwave electroacoustic devices applications. ThinSolidFilms, 2011, v.519, #19, pp. 6326 – 6329, 10.1016/j.tsf.2011.04.017

- Palatnikov, M. N., Sandler, V. A.,Sidorov, N. V., Makarova, O.V.,Manukovskaya, D.V. Conditions of application of LiNbO3 based piezoelectric resonators at high temperatures. Physics Letters A, 2020. V.384, #14, #126289, 10.1016/j.physleta.2020.126289

- Zhgoon, S.A., Shvetsov, A.S., Sakharov, S.A.,Elmazria, Omar High-Temperature SAW Resonator Sensors: Electrode Design Specifics. IEEE trans on Ultras Ferroel and Freq Contr., 2018, v.65, #4, pp.657-664, 10.1109/TUFFC.2018.2797093

- Kim, N.-In., Chang, Y.-L., Chen, J., Barbee, T., Wang, W., Kim, J.-Y., Kwon, M.-K.., Shervin, S., Moradnia, M., Pouladi, S. Piezoelectric pressure sensor based on flexible gallium nitride thin-film for harsh-environment and high-temperature applications. Sensors and Actuators A. 2020, v.305, 111940, 10.1016/j.sna.2020.111940

or something else.

  1. Under description of mechanism of piezoelectric effect it is better to use books like Auld, 1973 and for formulae line 84 use book Oliner, 1978.
  2. In the caption to Fig.1 (lines 115 and 119 refs.[13] and [14] should be [8] and [9], correspondingly, because before Figure 1 last ref. is [7].
  3. In line 129 ref [10] is not appropriate because it about microfluidic device not about FET.
  4. Check, please, refs.[11-14]. Are they appropriate here?
  5. Line 167: The Emnoman principle looks like Curie-Neumann’s symmetry principle. Give, please, appropriate reference.
  6. Title of item 3.2, lines 286, 403. Why do you use term “nanocomposites”? Look like it is simple “piezoelectric composites”.
  7. Add, please, at line 248 ref:

- Wu, J, Gao X.,Chen J.,Wang C.-M.,Zhang S.,Dong S.Review of high temperature piezoelectric materials, devices, and applications. Acta Physica Sinica, 2018, v. 67, #20, 207701, 10.7498/aps.67.20181091

  1. Add, please, at line 285 ref:

- Huang, S, Zeng, J,Zheng, L,Man, Z,Ruan, X,Shi, Xue,Li, G. A novel piezoelectric ceramic with high Curie temperature and highpiezoelectric coefficient, Ceramics International, 2020, v. 46, #5, pp. 6212-6216, 10.1016/j.ceramint.2019.11.089

  1. Lines 289-297, line 547 give, please more fresh references.
  2. Line 422, 743 give, please, appropriate Refs. for formulas.
  3. Line 457 correct grammar Li’s X.D, line 501 “piezoelectric”, line 503: remove crossed out text.
  4. Refs [55] and [56] are the same.
  5. When you talking about LGS (line 580) give, please refs.

-Sakharov S., Senushencov P., Medvedev A., Pisarevsky Yu. New data on temperature stability and acoustical losses of langasite crystals//Proceedings of the 1995 49th Annual IEEE International Frequency Control Symposium, 1995, pp. 647 - 652.  

- Kugaenko O.M., Uvarova S.S., Krylov S.A., Senatulin B.R.,  Petrakov V.S., Buzanov O.A., Egorov V.N., Sakharov S.A. Basic thermophysical parameters of langasite (La3Ga5SiO14), langatate (La3Ta0.5Ga5.5O14), and catangasite (Ca3TaGa3Si2O14) single crystals in a temperature range of 25 to 1000°C// Bulletin of the Russian Academy of Sciences: Physics, 2012, v.76, #11, pp. 1258 - 1263, 10.3103/S1062873812110123

  1. Item 7. Use instead “nanomaterial” “composite material”. Line 993 -Acoustic wave should be (AW). Lines 993, 995, 996: SAW is surface acoustic wave.

As for plate acoustic wave it is possible to add:

- Kuznetsova I.E., Zaitsev B.D., Joshi S.G., Temperature characteristics of acoustic waves propagating in thin piezoelectric plates// Proc. of IEEE Int. Ultras. Symp., 7-10 Oct., 2001, Atlanta, USA, V.1, pp. 157-160.

  1. As for development of sensors for temperature measurement it is possible to add ref.:

- Anisimkin V.I., Kuznetsova I.E. Selective Surface Acoustic Wave Detection of the Temperature of a Liquid Microsample// Journal of Communications Technology and Electronics, 64(8), 823-826, 10.1134/S1064226919080011

Author Response

Re: Nanomaterials (nanomaterials-1562597)

To the Reviewer:

Thank you for your reviewing on our manuscript “Nanomaterials for High Temperature Piezoelectric Applications” co-authored by Yanfang Meng et al. with provide enthusiastic, positive comments. We have revised the manuscript based on the your comments. We thank your comments/ suggestions and your efforts, which have helped to improve the overall quality of the manuscript.

The following are our replies (marked in blue) to the reviewers and modification to the manuscript (marked in red). We hope the revised manuscript meets the high standard of Nanomaterials.

We looked forward to hearing from you.

Yours sincerely,

Yangfang Meng

State Key Laboratory of Advanced Optical Communications System and Networks, School of Electronics Engineering and Computer Science, Peking University,Beijing 100871, China.  Center for Flexible Electronics Technology, Tsinghua University, Beijing, 100084, China.

Tel: 86-(10)62784622 

Round 2

Reviewer 1 Report

The mauscript has been sufficiently revised by the authors according to the given comments.